# Innovations in the T&L (Transport and Logistics) Sector during the COVID-19 Pandemic in Sweden, Germany and Poland

Monika Klein [1], Ewelina Gutowska [2] and Piotr Gutowski [1,*]

[1]  Management Institute, University of Szczecin, 70-453 Szczecin, Poland; monika.tomczyk@usz.edu.pl
[2]  Department of Economics, Jacob of Paradies University, 66-400 Gorzów Wielkopolski, Poland; egutowska@ajp.edu.pl
*   Correspondence: piotr.gutowski@usz.edu.pl; Tel.: +48-505-159-567

**Abstract:** Innovation is one of the most important factors stimulating the economy. It plays a special role in the Transport and Logistics (T&L) sector as it enables the acceleration of meeting needs process. During the COVID-19 coronavirus pandemic, many industries were and still are facing a tough economic test. The recession is also noticeable in transport, freight forwarding and logistics. However, how does this sector cope with the existing problems? Has the adoption rate of innovation been stopped in this sector? Do T&L developers see the potential of innovations and do they see them as a remedy and response to the pandemic threat? These issues have been thoroughly considered in the presented publication. The paper presents conclusions and selected results from a study on the adoption of innovations by companies in the transport and logistics sector during the COVID-19 coronavirus pandemic in Sweden, Germany and Poland. As many as three research hypotheses were adopted, which after being subjected to statistical fractional verification and evaluated substantively on the basis of the literature review and conclusions of research conducted, proved to be true. The aim of this paper was to verify the principles and determinants of innovation policy in T&L enterprises in selected countries during the pandemic crisis. Moreover, the paper contains an analysis of the entrepreneurs' experiences in the context of improving and developing their activities during economic crises, e.g., in 2008. It also presents the motivation and methodology of research. In addition to standard quantitative summaries, the authors conducted identification of correlations between the studied phenomena using the Cramer's V method and chi-square statistics. Obtained results allowed to better understand the processes taking place and to determine the general state and prospects of further innovation development in the T&L sector during the pandemic and ubiquitous restrictions.

**Keywords:** innovation during the crisis; innovation in the T&L sector; statistical fractional verification; statistical correlations

## 1. Introduction

Humanity faces many challenges today. In order to survive, it must adopt a strategy for the transformation of classical economic models into future-oriented sustainable, innovative and green economies that will provide opportunities for further civilization development and increased living comfort [1]. The decisions that will be made at this critical time will have their consequences in the years to come and will shape the economic and social reality in which we will exist.

The COVID-19 pandemic undoubtedly has a destructive impact on the economy of many countries. However, there are also new opportunities arising out of the radical and often dramatic changes. The word "crisis" derives from Chinese and literally means a combination of threat and opportunity [2].

The famous economist, Joseph Schumpeter, introduced the term "creative destruction" into the canon of socioeconomic terms, claiming that breakthroughs are most often made when the old order of things collapses, thus giving space for new, fresh and more effective perspectives [3].

According to the EU, economic success during and after the pandemic can be achieved by those who are research and innovation-oriented. Development and implementation of new breakthrough solutions and technologies have a direct impact on ensuring the resilience of manufacturing processes, accelerating economic and social transformation as well as stimulating all related sectors and clusters in a convergent way [4].

Historical context suggests that innovation-oriented companies not only function better during the crisis, but also benefit much more during the post-crisis period. Referring to the recent economic crisis, it can be seen that innovators at the turn of 2008 and 2009 achieved on average 10% better market performance than the competition. The difference was increasing in the following years to finally reach the maximum, i.e., 30% advantage, in 2012 [5].

However, it should be emphasized that not all innovations proved to be equally useful and beneficial. Product innovations survived the crisis much better, while the greatest weakness was revealed by innovations related to finances [6].

In the modern post-industrial world created by digital technologies, there are several key sectors that are critical for the proper functioning and development of economies. One of them is T&L (transport and logistics) [7,8]. It can be described as a highly creative sector, with great development potential and the ability to absorb modern solutions on a mass scale. The number of innovations in this sector, measured by the number of patents submitted, was characterized by annual steady growth (for example, in 2019 it increased by 2.6% compared to 2018) [9].

A well-functioning and efficient T&L is the key to success and competitiveness development. However, it is important for T&L to develop in a sustainable manner, thus ensuring the possibility and chances of a comfortable existence for future generations. For this purpose, it cannot use the planet's resources in an uncontrolled way or pose a serious ecological threat to the planet (transport alone is responsible for nearly a quarter of the world's greenhouse gas emissions) [10]. Future mobility must be based on the optimized use of all means of transport, sometimes translating the economic profit over the social and environmental one. The European Union, understanding the threat posed by this state of affairs, and bearing in mind the close relationship between T&L and economic growth, prosperity and global trade, has decided to adopt strategies to stimulate solutions that lead to sustainability. The most rapid way to achieve this is through policies that support deployment and the diffusion of innovation [11].

Many of the innovations are based directly or indirectly on the development and diffusion of digital technologies, and the T&L sector has been at the heart of this revolution for many years [12]. The ICTs did not suffer during the pandemic because they are primarily based on the human mental potential and this has not changed or degraded. On the other hand, the demand for all ICT solutions, both in the hardware and software sphere, has increased. This may lead to acceleration of digital technologies. However, it is not clear whether this development potential will result directly in stimulating innovations that can support the economy or will be limited only to the social zone [13]. Economically weak companies may be reluctant and very cautious about new investments and costly innovations. In this case, changes in the structure and hierarchy of many markets should be expected, which may be distorted by dynamic innovative enterprises and companies that have maintained their financial standing and adopted a pro-development strategy.

It is worth asking a question: can the crisis related to the COVID-19 coronavirus pandemic have an unexpected effect and become a catalyst for innovation? Many of the main suppliers of modern technologies have financial resources and advanced studies on new methods to increase the efficiency of industry, living comfort and environmental protection. Therefore, it is doubtful that they will easily give up on further research, especially since they have strategic resources to meet them. Moreover, the threat to health and life has again increased public pressure on environmental issues, which may force key sectors of the economy such as T&L to adapt more quickly to the principles of environmental neutrality. This is only possible through adoption and support for innovation. On the other hand,

pandemic time indicates uncertainty and a sense of threat. Many investors are skeptical, and the capital is invested in what are perceived as safe and stable spaces such as precious metals and works of art. It is important to remember that the crisis affected the surprised world at the moment when innovation boomed, stimulated by public finances. It may turn out that the necessity of redirecting funds and allocating them to more current needs may lower the determination of those in power to support the innovation development [14].

However, the economic crisis that affected the entire world between 2008 and 2014 teaches humility and prompts moderate optimism in the context of hopes that the economic collapse related to the pandemic will become a strong stimulator of innovation development. Data collected at the time showed that the crisis significantly reduced the number of companies in Europe willing to increase their investment in innovation from 38% to just 9%. This does not mean that the interest in innovations decreased, on the contrary—it intensified in the last phase and right after the crisis. Nevertheless, at that time mainly cheaper solutions were sought. On the other hand, 9% of companies that increased their pro-development outlays during the economic turmoil did so very dynamically [15].

In this paper, the authors will analyze and evaluate the current situation related to innovation in the T&L sector among companies from Sweden, Poland and Germany. On the basis of literature study, the situation before and during the pandemic will be presented and scientific hypotheses will be formulated. In the following part, the methodology of conducted research and its results will be presented, extended by comments and observations as well as the authors' opinions. The whole will be concluded with a summary, in which the reader will find an answer as to whether the accepted hypotheses turned out to be true or false, and a synthetic summary of the whole argument and conclusions from the research.

The authors have focused on the research topic addressed as it is extremely relevant to the global economy as a whole. The size of the global logistics market may be indicated by the fact that it is expected to nearly double in value from just over $7.5 trillion in 2017 to nearly $13 trillion in 2027 in just a decade [16]. This is because there is an obvious convergence between the logistics market and the ICT sector and e-commerce in particular. Such a state of affairs results in accelerating the development of logistics while maintaining the principles of its sustainability becomes one of the main global imperatives. One of the factors influencing this acceleration is the effective adoption of innovations. The authors adopted as the aim of their research to investigate how this adoption progresses during the economic downturn caused by the omnipresent pandemic threat. The results and conclusions of conducted research may be very relevant and useful when implementing countermeasures in case of similar crises and provide a guideline for T&L companies and enterprises.

## 2. Theoretical Analysis and Research Hypotheses

### 2.1. Literature Review

The pandemic time, which is certainly a tragic and traumatic period in the history of modern civilization, had a very strong impact and influence on economic life. Many researchers and economists have taken advantage of this difficult time to conduct research aimed at creating more resilient and flexible economic models based on innovation, which can be seen as a weapon to fight COVID-19 [17,18]. Research in this context has also focused on the T&L sector, but has addressed its different aspects and focused on different regions of the world (Table 1).

**Table 1.** Examples of studies on the T&L sector during the COVID-19 pandemic.

| Title | Study Details | Key Findings |
|---|---|---|
| COVID-19 Impact on the Logistics Industry [19] | Surveyed entity: a company based in Dubai that offers its services and trades throughout the Middle East. | The negative impact of COVID-19 was observed in almost every aspect of the company's operations. The company could not meet timely deliveries which ultimately led to a decline in its revenue. |
| Impact of COVID-19 on the supply chain industry [20] | Impact of COVID-19 on the health of supply chains using Nigeria as an example. | The manufacturers were not able to meet the market demands due to inability to purchase raw materials on time. This situation led to increased inflation and depleted supply of goods. In order to improve the situation, some of the companies reached for innovative solutions usually based on ICT, which proved to be an effective solution. |
| Impact of COVID-19 on transportation and logistics: a case of China [21] | Quantitative research on the impact of pandemic on T&L sector in China. The research focused on three spaces: land, sea and air logistics. | The study results proved that COVID-19 significantly and negatively affected the air and land logistics sector. However, no statistically significant relationship was observed with the maritime logistics sector. |
| Moving towards "mobile warehouse": Last-mile logistics during COVID-19 and beyond [22] | Literature study and analytical models. | Innovation related to the derivation of mobile storage can be an effective tool to counteract the difficulties caused by COVID-19. |
| The Impact Of COVID-19 On Logistic Systems: An Italian Case Study [23] | Case Study—quantitative and qualitative study of companies operating in the logistics industry in Italy. | All activities related to the introduction by companies of protections against COVID-19 had a significant impact on its financial health. During the pandemic, customer preferences changed which also affected logistics companies. |
| Digital Transformation in Latin American and Caribbean Logistics [24] | Secondary data, literature review. South America and the Caribbean. | The COVID-19 pandemic became a catalyst for the digitization of trade and logistics. |
| Fast Forward. Rethinking supply chain resilience for a post-COVID-19 world [25] | Quantitative and qualitative research. 1.000 surveys targeted to consumer industry entrepreneurs and in-depth interviews with selected retail chain executives. The survey was conducted in 11 selected countries around the world. | The supply chains of the vast majority of organizations did not survive the test of coronavirus pandemic. Based on this experience, companies have taken numerous initiatives to increase their supply chain flexibility, but they do so on sustainable terms. |

Source: own elaboration.

It is important to note that the main goal of T&L companies is to store and distribute goods efficiently through flexible supply chains [26]. However, the emergence of the pandemic threat has put the entire logistics system to a very demanding test [27]. Operators have comprehended an uneven fight against numerous constraints (such as restrictions) and other problems (e.g., staff shortages or availability of goods and services) in order to maintain the entirety of meeting their customers' needs [28,29]. Some researchers have realized that innovation is the most effective way to achieve this goal and have focused their interest and research potential on it [30]. Despite the relatively large number of publications in the sector, according to the authors, there is still a lack of sufficiently insightful primary research, which is not a case study of one company (such as [31]), one country (such as [32]) or based on secondary data (such as [33]). It should be emphasized that the conclusions presented in the paper are based on research conducted in three countries and concerning as many as 1597 economic entities, which is very rare in scientific papers. This makes the results obtained extremely valuable and have a very high credibility level. However, it cannot be ruled out that similar studies will appear in the near future, as the pandemic is a phenomenon that began suddenly and is still ongoing.

*2.2. Innovation and T&L Sector before the Pandemic*

The main forces responsible for T&L industry development before the pandemic include digitalization, shifting the international commercial center of gravity to the Asian region, evolution and modernization of production processes through the development and dissemination of computer skills and software (e.g., Artificial Intelligence (AI), Internet of Things (IoT), Big Data Analytics (BDA), Blockchain, business intelligence (BI), 3D printing, intelligent sensors, etc.), development of new trade forms and increased consumption (e-commerce and availability of goods from almost anywhere in the world) as well as improvement and development of machines [8]. All these forces are highly innovative-creative and innovative-absorbing, but digital technologies seem to be the overriding and dominant creator. However, it should not be forgotten that innovation is not just a new technology, but rather a set of factors and actions leading to the improvement, creation and implementation of processes and a more efficient satisfaction of needs.

In 2018, Blockchain was considered a breakthrough innovation in the T&L sector. The following places were taken by Artificial Intelligence, robotics, independent vehicles and drones. All these innovations are highly correlated directly with technological development. It was the technological transformation that was considered to be the biggest challenge for the T&L sector, taking first place before other factors: meeting the growing demands of clients, implementing innovations, regulations and competition. The biggest threat from the point of view of a single company is a very competitive and dynamic market. In the context of innovation implementation, on the other hand, the reluctance to change, high risk associated with the possible unprofitability of investments, limited financial resources and lack of creativity, ideas, determination and visionary leadership are considered significant barriers. These problems correlate with the declaration of holding an own implementation team, as in most companies there is no such a cell or specialized department (there is a very large discrepancy between companies familiar with technical innovations and those that do not have a lot of knowledge in this field, in the context of awareness regarding the need for continuous development and improvement; in the former, specialized cells for research and analysis of innovation implementation possibilities appear in half of the cases, and in the latter, a similar cell appears only in every fourth company) [34,35].

Although discussing some T&L spaces, e.g., warehouse management, it was said that "the future has already come" [36] as solutions that seem to be completely autonomous and very effective become the standard, there were sectors affected very much by a lack of innovation and digitalization. One of them was supply chains, where the lack of full control over the flow of goods due to insufficient and incompatible digitalization among partners resulted in the generation of significant costs and the impossibility to achieve many benefits, such as: increased customer experience, quicker product launches, ability to react quickly against emerging threats, creating loyalty ties with the client, etc. [37].

In the years before the crisis, a spectacular change in the perception of T&L service providers regarding meeting the growing demands of their customers can be seen. The information society, where access to knowledge is not an obstacle, has become more conscious and thus more demanding. Research shows that in 2017, in the context of the most important factors influencing the functioning of T&L, customer expectations and consumer marketing were only ranked fifth with 9.5% of all votes. In 2018, they were already in third place with 20.2% of votes [38]. In the following year 2019, the attention of entrepreneurs in this aspect increased even more, eventually reaching 37% [39]. The evolution regarding the change of thinking in relation to the needs for functional conditions of T&L companies in the era of ubiquitous and galloping innovation has been presented in Figure 1.

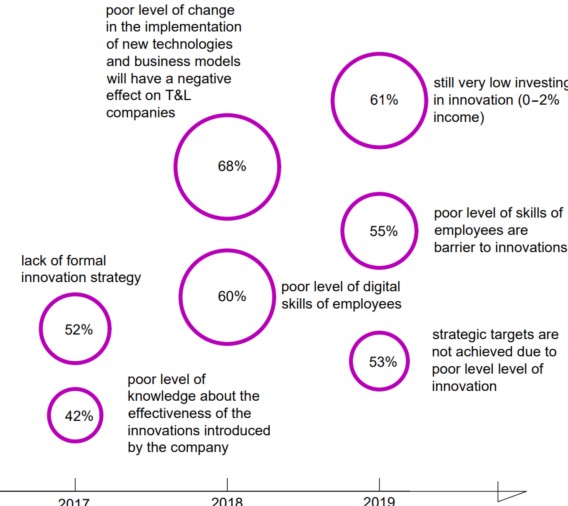

**Figure 1.** Shortages and weaknesses of T&L companies in the three years preceding the pandemic. Own elaboration on the basis of: [5,35,40,41].

Companies in the T&L sector have sufficient knowledge of market trends and their weaknesses. They are also able to identify the causes of their failures to implement and develop new solutions, but do not take radical steps in order to change this state of affairs. They treat innovations as an investment of very high risk and doubtful profitability rather than as an opportunity and tool to improve their market situation. This is evidenced by very low development outlays presented in Figure 1.

### 2.3. Innovation and T&L Sector in Pandemic Time

There are extremely few up-to-date and reliable data available on the current state of innovation in the T&L sector. This is undoubtedly related to the fact that observed phenomena are taking place and shaping in real time, which significantly hinders literature study. Nevertheless, there are exceptions. The McKinsey Institute has conducted a survey among the management staff of companies, which shows that 90% of the respondents are of an opinion that the coronavirus pandemic will change the way business is organized for the next 5 years. 85% of the respondents believe that customers' preferences, habits and tastes will change. Unfortunately, almost 80% of companies do not have enough knowledge and resources to effectively stimulate their further development and adopt modern solutions and innovations. At the same time, 2/3 of the respondents are aware that the year 2020 is likely to be a breakthrough in their careers. The decrease in interest as regards innovations during the pandemic was reflected in all industries and all countries surveyed. The medical and pharmaceutical industries were an exception. The companies focused on defensive actions consisting of: supporting the core business, applying proven mechanisms, seeking savings and giving up ground-breaking decisions in anticipation of stabilizing the situation. Meanwhile, a more appropriate attitude should be: keeping track of changing customer preferences and adapting to their new requirements, responding quickly to the fluctuating marketing environment, assessing own capabilities and redistributing the company's resources in proportion to the current demand and burden of individual departments, creating a vision of the company's existence in the post-COVID-19 reality. In order to maintain a competitive market position, business organizations must not remain in stagnation because the conditions for their continued operation have changed in many countries. For example, the regulatory context, availability of natural resources and labor force, money purchasing power, etc. have changed. At the same time, new opportunities emerged that never existed before [5].

The T&L sector has always been characterized by great variability. However, referring to the times before pandemic, it could not be called unpredictable, but rather highly developed. Driven by market trends, it adopted and used new technological solutions at a

very high pace, expanding its complexity and convergence with other industries. Given the total amount of Research and Development (R&D) spending in 2018 and 2019, the T&L ranked fifth in the world in this respect, using 3.8% of total funding [14]. Therefore, the fact is that there is a very strong link between the financing of T&L development and the global GDP, which decreased sharply in 2020. The most noticeable recession took place in the second quarter of 2020 and since then global finance has been slowly stabilizing and making up for losses thanks to the stimulation policy (Figure 2) [42].

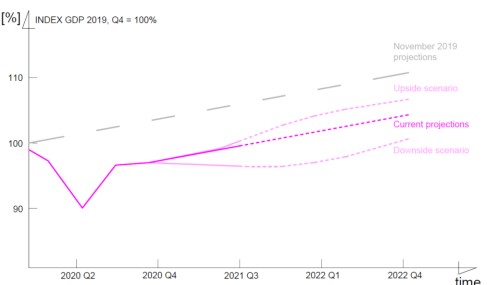

**Figure 2.** Global GDP. Forecast. Own elaboration on the basis of: [42,43].

*2.4. Research Hypotheses*

The literature studies and conclusions that can be drawn from the available secondary data raise more questions than answers about the condition of T&L sector during the pandemic. Did it maintain the absorption rate of innovative solutions? What kind of innovations were implemented during this period? Did decision makers see an opportunity in innovations and treat them as a remedy for the COVID-19 pandemic? Or maybe on the contrary—they are cautious, accumulate capital, look for savings and wait what the world will look akin to after a pandemic crisis?

In order to at least partially answer these and other questions related to the issues under consideration, the authors have planned and conducted surveys that have covered several EU countries. They also adopted three research hypotheses, formulated on the basis of available knowledge and their experience. The first hypothesis is consistent with the observations made by the McKinsey Institute [5] and assumes that most of the companies in the T&L sector are aware of the fact that stimulation and implementation of innovative solutions at the critical time of pandemic is a great development opportunity for them, giving the prospect of gaining a much better market position in the future (H1).

Hypothesis two is closely related to the first hypothesis as it is a kind of its elaboration. It is based on the assumption that although most T&L companies have the key knowledge and resources necessary to implement innovations, they do not actually pursue them due to the unpredictability related to the COVID-19 pandemic (H2). It should be added that H2 can only be considered if H1 proves true. H2 is a contradictory hypothesis, but it is located in strictly defined conditions and conflicts with logical and consistent action. In view of the above, the authors decided to make an exception and include it in their paper.

The third hypothesis is related to the fact that the "lock down" affected and still affecting the economy has forced companies to reorganize their structures and adapt to the prevailing situation in order to sustain their operations. Hence the conclusion that: during the COVID-19 pandemic, the innovations implemented by T&L companies are mainly organizational innovations (H3).

In summary, three hypotheses were developed:

**Hypothesis 1 (H1).** *Most T&L companies are aware that the stimulation and implementation of innovation during a pandemic crisis can bring very high returns and competitive advantages after the pandemic is over.*

**Hypothesis 2 (H2).** *Despite their knowledge of the enormous potential investment in innovation during a pandemic crisis, most T&L companies do not pursue it.*

**Hypothesis 3 (H3).** *During the COVID-19 pandemic, the innovations implemented by T&L companies are mainly organizational innovations.*

### 3. Materials and Methods

The paper is based on a literature study and own research. Literature analysis concerned mainly different types of secondary data sources such as reports or databases. Moreover, papers (both in electronic and traditional versions) and books were used, although the latter treating the subject matter were found very little, which certainly results from the topicality of this subject.

Our own research was conducted using only the potential of a three-person research team. The research results included in the publication were not in any way financed from funds of any projects or other institutions. In order to meet such a large challenge at minimum cost, the authors used a large network of contacts (both professional and private), asking for help in disseminating the survey. They also used the power of modern media—social networking sites, which were used to establish relations with specialized thematic groups.

The survey was developed by the authors and evaluated by three independent professors, working at various universities and being recognized experts in the field. After collecting opinions, the survey was verified and preliminary pilot-testing was conducted on 50 Polish companies. It took place in July and August 2020. Then the results were statistically processed. The whole process was carried out without any problems and the obtained data turned out to be consistent and usable in scientific work.

The data presented in some graphs or diagrams may not add up to 100% due to the assumed rounding of values or the possibility of giving more than one answer. The authors adopted the company's country of origin and its size measured by the number of employees as descriptive variables.

The subject matter was related to very current and new phenomena. Although already on January 30, 2020 the World Health Organization recognized coronavirus as an international public health emergency [44], the number of cases in Europe started to increase rapidly only in mid-February 2020 [45], which certainly started to affect the behavior, dispositions and decisions of business representatives, including the T&L sector.

Due to the relatively high effectiveness, global character and time of international epidemiological threat, the authors decided to conduct studies using CAWI (Computer-Assisted Web Interview) method. In a few cases the CATI (Computer Assisted Telephone Interview) and PAPI (Paper & Pen Personal Interview) techniques were additionally used (mainly in Poland, but in very few cases). The research started at the beginning of October 2020 and lasted until 10 December 2020.

Surveys were addressed to employees of T&L companies in three countries: Germany, Sweden and Poland. Statistical data show that the total number of T&L entities in these countries was 289,797 [46]. However, these data do not come from 2020 and are not up-to-date. In order to ensure the appropriate quality and desired level of results reliability, the statistical community was eventually set at 380,000. This assumption allowed to calculate a representative sample of 384 (with a maximum error of 5% and an estimated 95% confidence level).

The following equation was used to calculate the study sample (Equation (1)):

$$n = \frac{u_\alpha^2}{4d^2},\tag{1}$$

The value of $u\alpha$ statistic for the probability assumed, as read from the tables of normal distribution, was 1.96.

The survey was conducted among 1597 companies, 840 of which stopped to answer only the first two questions declaring that in 2020 they did not implement or develop any innovative solution in their organization. A further 40 surveys were rejected due

to incorrectly filled in document or grossly incomplete answers. Therefore, the basis for analysis was 717 surveys, which was almost twice (1.87) the required research sample.

The research was made using Facebook and Instagram, company and private e-mail addresses and contacts on LinkedIn. Depending on the preferences reported by the respondents, questionnaires were filled out and sent in different ways. From a form filled out electronically in Google Forms, to a completed, scanned and sent survey in a format such as PDF. These different forms and individualized approach were very labor intensive, especially with such a large sample, but at the same time it allowed for the collection of a very large number of correctly completed forms.

It was decided to carry out statistical detection of regularities in the field of correlation of phenomena. The statistics $\lambda^2$ and the V-Cramer method were used as specialist tools. Authors chose these solutions, because the results appeared more effective than another statistical tools as Txy Czuprow, C—Pearson, or Ø Yule. The number of columns and rows in the association table is independent using V-Cramer method that allows to measure the relationship between values of variables express on nominal scales. The results can achieve values from the range [0, 1]. If the obtained result comes closer to unity, that means there is a very strong relation between qualitative variables. If the outcome brings closer to zero it means lack of the independence of the analyzed characteristics. The level of the significance was adopted to ≤0.05 for chi test. For the V—Cramer's coefficient the following thresholds were adopted to determine the strength of the relation: ≤0.25—no significant relation; (0.25;0.35>—a weak relation; (0.35;0.45> a relation of moderate strength; (0.45;0.55>—a relation of great strength; above 0.55—a relation of very great strength.

### 4. Who Were the Respondents?

Statistical analysis focused on the T&L sector entities which implemented innovative solutions in their structures during the pandemic. Although companies from the three countries were addressed in the project, 3% of the total expressed the opinion that they are international organizations. The percentage of respondents from particular countries was 39% from Poland, 32% from Germany and 26% from Sweden, respectively (Figure 3).

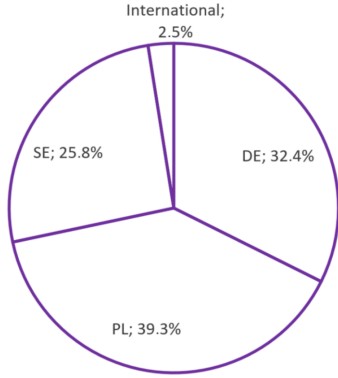

**Figure 3.** Nationality of respondents. Source: own elaboration.

In terms of enterprise size, the research sample included large companies with more than 250 employees, 4% of the total, medium-sized companies with 50 to 250 employees, 27% of the total, small companies with 10 to 49 employees, 54% of the total (the most numerous group) and micro companies with less than 10 employees—15% of the total (Figure 4). Large companies are mainly Swedish companies—35.5% of the total. The largest number of medium-sized companies came from Poland—41.3%. Among small and micro companies in the context of the country of origin the proportions were very evenly distributed. The exception were international companies—these can be mainly attributed to the group of large and medium companies.

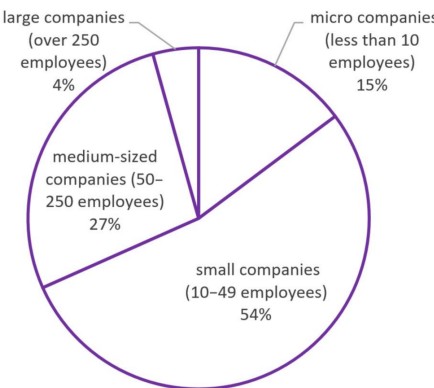

**Figure 4.** Size of the surveyed companies. Source: own elaboration.

Among the audited entities, the dominant were those that have been operating in the T&L sector for more than 3 years (from 3 to 10 years—41.3% and over 10 years—42.7%; total: 85%). The companies from Sweden and Germany that took part in the survey had a relatively long tradition and extensive experience. The situation with respect to Polish companies was slightly different—the most numerous group were organizations operating on the market for 3 to 10 years. The number of young companies was also significant—28.3%. Combining the results of business age declarations with the company's size, an interesting trend can be observed, from which it can be concluded that larger companies stay on the market longer.

## 5. Results and Discussion

During the period of coronavirus threat, the majority of companies gave up on innovation. This was expressed by 53.9% of the respondents. The most important reasons for this decision were: lack of funds (20.3% of the total), COVID-19 (20% of the total), lack of adequate personnel knowledge and skills (14.9% of the total), lack of knowledge about funding opportunities (14.8% of the total), lack of sufficient support from state institutions (11.2% of the total), lack of need for innovation (8.9% of the total) and other reasons (9.9% of the total). This hierarchy was different in terms of the origin of enterprises. For German entities, the most important factor was COVID-19 (28.2% among German companies) and the least important was lack of knowledge about financing possibilities (6.7% among German companies). For Polish respondents the biggest barrier was the lack of funds (27.8% among Polish companies), and the least numerous opinions were given out in defense of the position of lack of the need (only 3.7%). In Sweden, economic organizations, similarly to Germany, adopted COVID-19 as the most important factor (28.3% among Swedish companies) and the lack of funds was perceived as the least problem (7.1% among Swedish companies). Entities that have described themselves as international also took the view that COVID-19 was the main obstacle to innovation (45.5% among international companies). Data analysis revealed the existence of a statistically significant relationship between the reasons why firms did not introduce innovations in 2020, and the country where those firms operate.

The results obtained are consistent with the research results and observations made by other researchers. Lack of appropriate knowledge and skills as well as lack of funds repeatedly appear as the main barriers to innovation development [43,47,48]. Another very important obstacle reported by companies was the development of coronavirus pandemic, which affected the economic life of the whole world. The COVID-19 represents a serious threat to innovation, which should be protected from its effects and additionally stimulated by government programs [49]. The opinions of companies located in different countries differ slightly. Those companies that operate in highly developed countries (Sweden second and Germany ninth in the world according to the Global Innovation Index (GII) in 2020 [14]) have less problems with financing innovation than entities in developing countries (such

as Poland—38th in the world [14]). The values of GII index also correspond to the number of innovative economic organizations in different countries (Figure 5). The least modern solutions are adopted by the T&L sector in Poland—40.2%, a little more in Germany—45.4% and the most in Sweden—57.5% and by international companies 78.3%.

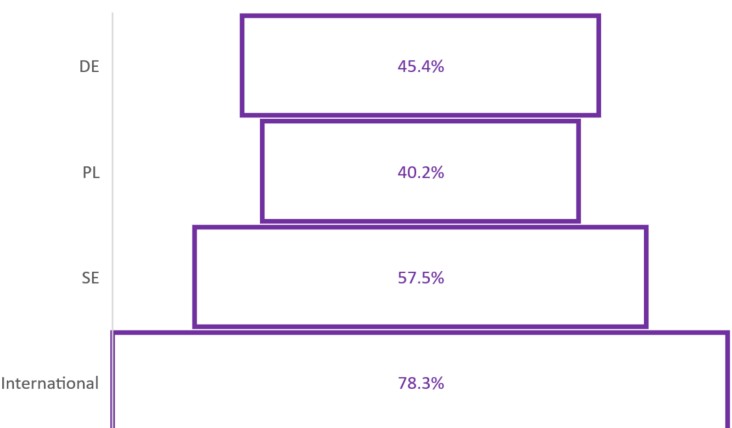

**Figure 5.** Percentage of innovative companies in the T&L sector in individual countries. Source: own elaboration.

The number of innovations implemented in companies exceeded the value of "1" and was 1.423. This means that if an entity reaches for a new solution, it is not a one-off action. The indicator achieved different values for data collected in different countries. The lowest value was recorded for Polish companies: 1.106, followed by German companies: 1.129 and Swedish companies, respectively: 2.189 and international: 2.278. In 2020, the most frequently implemented innovations were organizational innovations. They constituted as much as 55% of the total and were the most numerous groups regardless of the company size and country. The next places were taken by process innovations—17.8%, product innovations—16.3% and marketing innovations—10.9%.

It can be assumed that such a large discrepancy results from and is caused by the COVID-19 pandemic. Many of the companies had to change to a remote mode of operation, adopting different organizational models. Certainly, many of them were not breakthrough from the novelty point of view, but from the innovation point of view they were new within the organization. Examples of organizational innovations can be the first launch of a quality management system or a new model of employee rights, decision-making and tasks.

A total of 61.1% implemented innovations were interrelated, thus the introduction of one modern solution was a consequence of introducing another innovation. The company is a mechanism of mutually complementary and cooperating elements. Introduction of a change in one space may force the necessity to change another element in order to fully use the new potential or maintain the desired level of compatibility. The respondents were of an opinion that among related innovations, the first one was most often a process innovation (53.7%) or that the solutions were introduced at the same time (28.2%).

Use of V-Cramer index gave a proof of existence of a statistically significant connection between the first implemented innovation and the country where firm operates. The reason for this may be different levels of particular branches of economy, which, at the same time, require different innovative solutions and particularly the sequence of their implementation. Additional reasons may be related to strategical goals and resources of a given country, development of infrastructure as well as politics. Despite the differences, it is possible to notice some similarities. These, in turn, may result from general digitalization and informatization, to which almost every area of economic and human existence is unconditionally subordinated. Statements on the sequence of implementation of particular types of innovations in the view of a company's country of origin have been included in Table 2.

**Table 2.** The first implemented innovation and the country where the firm operates.

| Innovation | DE | PL | SE | Inter. |
|---|---|---|---|---|
| products related innovation | 27.8% | 26.3% | 1.9% | 44.4% |
| organizational innovation | 11.1% | 0.0% | 1.9% | 0.0% |
| marketing related innovation | 0.0% | 5.3% | 0.0% | 0.0% |
| process related innovation | 33.3% | 36.8% | 63.1% | 22.2% |
| they were introduced at the same time | 22.2% | 21.1% | 30.1% | 33.3% |
| difficult to define | 5.6% | 10.5% | 2.9% | 0.0% |

Source: own elaboration.

The most frequently indicated motivation for the decision to reach for innovations turned out to be the coronavirus pandemic—31.7% of responses (Figure 6). Almost as important was the competition pressure and customer requirements. Referring to the research results contained in the report "Supply chain innovation study" [40] a very large identity can be observed—according to the respondents, the most important factors that prompted them to modernize their companies were growing customer requirements and competitive environment. In this particular case, there appears also a significant statistical dependence in relation to the country in which the organization operates. The distribution is shown in Figure 6.

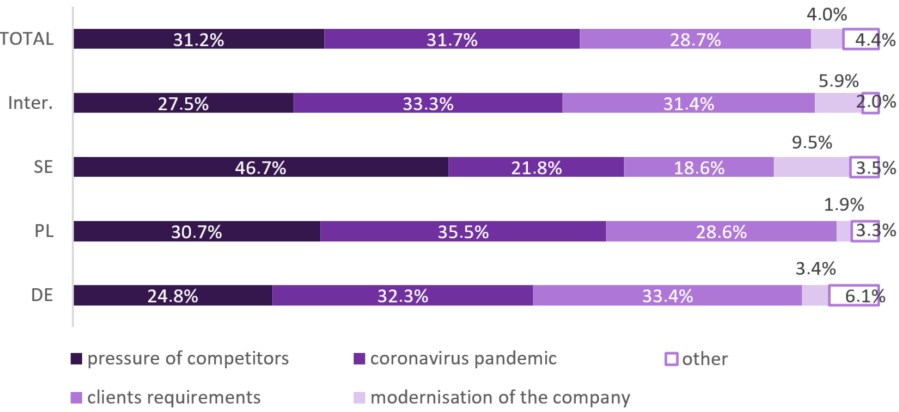

**Figure 6.** Reasons for implementation of modern solutions. Source: own elaboration.

The vast majority of companies evaluate the implemented innovations positively (Figure 7). 75.8% of the respondents stated that thanks to the innovations they achieved good or very good results. Slightly more than 16% did not notice the change, while less than 8% noticed the negative impact of implemented changes. The profile of the latter can be identified as mainly small companies from Poland. The reasons for this state of affairs can be seen in several factors. First of all, it is worth remembering that in Poland the main source of financing innovations are the company's own resources [50]. For small enterprises it is a heavy burden and often a strategic decision, after which they expect quick results. However, the effects of innovation implementation require some time, which causes a quick discouragement. If the financing of an innovation is provided by funds that do not belong to the company, unclear, lengthy and complicated procedures and bureaucracy pose a problem [51]. This makes the implemented solution at least partially perceived and evaluated through the prism of granting funds process.

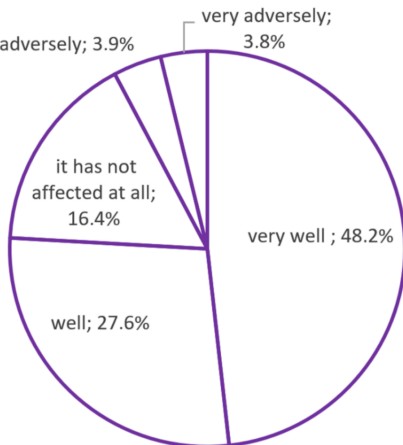

**Figure 7.** Evaluation of implemented innovations. Source: own elaboration.

There is a correlation between the evaluation of the effects of innovation and the location of the company. Optimistic approaches predominated in most countries in which the survey was conducted, i.e., in Germany, Sweden and among entities with an international profile. The exception to this is Poland, in which a moderate optimism was noted.

Innovative companies from the T&L industry during the pandemic mostly decided to conduct R&D. This was stated by 56.4% of the respondents. Among them, it is possible to distinguish between more and less active ones. As a consolation, the companies that declared to carry out six or more works constituted as much as 28.1% of the respondents. In comparison with the company's size and country of operation, the leader profile can be identified as a large Swedish or international organization.

The R&D is driven by the company's current needs. Such work is a prelude to innovation, as it allows to better understand own needs and opportunities, determine risks and possibilities, learn about the marketing environment and identify strengths and weaknesses of the business. It happens that they are an innovation alone or directly lead to an innovative solution. If this is not the case, they still have a positive impact on the organization's level of awareness, thus it can effectively and efficiently seek outside help and create realistic and thoughtful scenarios for the implementation of modern solutions, which significantly increases the chance of obtaining outside financing.

Innovations have an extremely important feature of driving economic growth and relatively high resistance to economic crises. The potential for innovation can be improved through planned and well thought-out support for R&D and creation of cooperation networks between business, industry and scientific centers and institutions [52].

The development of T&L constitutes an important point in the development strategies of many countries and international institutions. It is polarized with the development of innovations and may even be dependent on them in the future, because there is a steady increase in environmental policy and only innovations can lead the T&L sector to neutrality in this field [53].

The authors of this study, guided by the need for research and to identify as many causal interactions as possible, have identified five potential forces that may influence a company's decision to undertake R&D efforts:

- market research regarding future consumer needs or sales of products—33%,
- construction of models and prototypes of future products and their testing—21.8%,
- examining the current state of a given industry and predicting future technological developments in this industry—20.5%,
- supplementary research, patent or licensing—9.9%,
- other—7.8%.

Funding for these projects came primarily from two sources: government funds supporting companies during the coronavirus pandemic—23.8% of responses, and the

company's own funds—20.9% of cases. The data prove that there is a strong link between innovation and R&D. In both cases, the desire to meet the growing expectations and needs of clients plays a very important role, with R&D focusing more on detecting these needs in a fluctuating market environment and how to meet them, and innovation being a specific tool for real and effective action in this matter.

Enterprises recognize many benefits of R&D. Those who decided to choose it, mentioned most often as an advantage: reduced labor costs (45.64%), increased safety of employees (40.15%), staying on the market (35.66%) and increased competitiveness (30.17%). The list of perceived benefits has been presented in Figure 8.

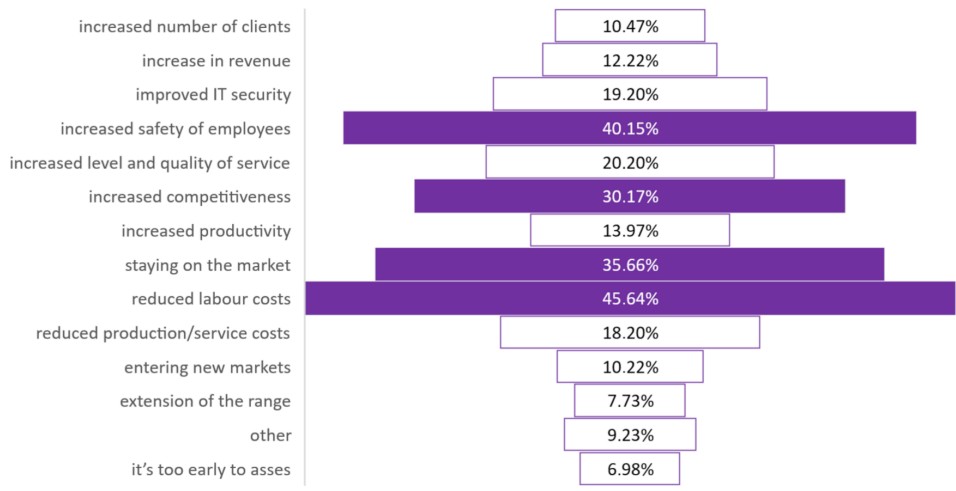

**Figure 8.** Advantages perceived by T&L companies from R&D. Source: own elaboration.

With the outbreak of the coronavirus pandemic, the global social and economic order has been significantly disturbed. This situation also applies to development activities and innovation. It is predicted that it will mostly have a negative impact on many economy sectors, but there will be such spaces that due to a direct association with the pandemic or a high involvement in its mitigation, will receive increased funding, which will accelerate their development and contribute to the development and implementation of numerous innovative ideas [5,54]. It is important to remember that the study concerned the pandemic period, during which the supply of easily accessible governmental (but not only) financial resources has increased, primarily aimed at all initiatives involving consequence reduction, contraction and the fight against coronavirus. Therefore, it can be assumed that the highest indications presented in Figure 8 are the result of this support policy and the emphasis would have been different in the period preceding COVID-19.

The surveyed entities are conscious enterprises, perfectly familiar with the rules of market functioning and the complexity of macroeconomic phenomena. This is evidenced by the declarations that have been illustrated in Figure 9.

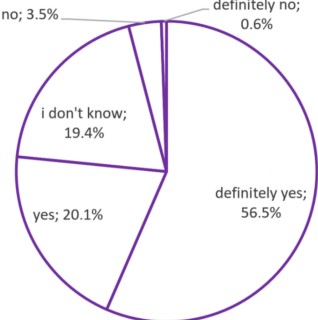

**Figure 9.** Question: is pandemic a critical time for innovation? Source: own elaboration.

As much as 76.6% of the surveyed entities were of an opinion that during the current crisis it was necessary to invest and implement new solutions, of which 56.5% reported absolute certainty. Less than 20% declared that they did not have enough knowledge to answer this question and only 4.1% of the respondents claimed that this is not the right time for such actions.

Authors use V-Cramer method to detect six correlations between studied phenomena. It is presented in Table 3.

**Table 3.** Correlation of phenomenon.

| Correlation of Phenomena | Strength of the Relation | Value of the V-Cramer Indicator |
|---|---|---|
| Correlation between: motivation for implement the innovations, and the country where this firm operates. | great strength | 0.673 |
| Correlation between: the subject witch make a decisions about implementing innovations, and size of company. | weak | 0.332 |
| Correlation between: the first implemented innovation and the country where this firm operates. | weak | 0.328 |
| Correlation between: the reasons why firms did not introduce innovations in 2020, and the country where these firms operate. | weak | 0.267 |
| Correlation between: the influence of implementation of innovations on the condition of company and the country where firm operates. | weak | 0.266 |
| Correlation between: the plans of company to use e-work after the SARS-CoV-2 pandemic and size of company. | weak | 0.26 |

Source: own elaboration.

The strongest correlation was found between the motivation for innovation and the company's country of origin. The specific results are presented in Figure 6. In this case, the value of the Cramer's V coefficient was as high as 0.673, which is the highest result in the entire study. Unfortunately, the method used does not permit the determination of the directionality of the identified relationship, but only to give an opinion on its strength, which in this case should be described as very high.

While in the case of Poland and Germany, the opinions of respondents were very similar, in the case of Sweden the economic motivation was by far the most important. Looking at constantly updated data on an interactive map developed by the Johns Hopkins University [54] concerning COVID-19 pandemic, it is possible to easily notice the huge disproportion in relation to the number of deaths in the discussed countries. Taking into account the status as of 26.02.2022, in Germany there were 122,639, in Poland 111,277 and in Sweden 17,142 victims of SARS-CoV-2 [55]. Due to the differences in population, it is worth introducing a simple auxiliary indicator (T) equal to the ratio of population and pandemic deaths. After performing the calculations, it was found that: $T_{SE} = 603$, $T_{DE} = 679$, $T_{PL} = 341$. The results obtained should be interpreted as follows: by the end of March 2022, one person out of every 603 citizens has died due to COVID-19 in Sweden, one person out of every 679 citizens in Germany and one person out of only 341 citizens in Poland. Poland's performance is by far the most unfavorable. The situation was almost twice as good in Sweden and Germany, respectively. The analysis performed is consistent with the study results. It was in Poland that the pandemic was the main motivator (35.5%—Figure 6) for the introduction of innovations, many of which involved innovative ways of protecting life and health.

Out of the weak correlations detected in the statistical analyses which are listed in Table 3, two are yet to be discussed. These are correlations between: the subject which makes a decision about implementing innovations and size of company, as well as correlations

between: the plans of company to use e-work after the SARS-CoV-2 pandemic and size of company (Tables 4 and 5).

**Table 4.** Correlation between: the subject which make a decisions about implementing innovations and size of company Correlation of phenomenon.

| | Micro Companies (less than 10 Employees) | Small Companies (10–49 Employees) | Medium Companies (50–250 Employees) | Big-Sized Companies (over 250 Employees) |
|---|---|---|---|---|
| the owner | 62.89% | 14.13% | 15.68% | 0.00% |
| the management makes such decisions itself (management board) | 15.46% | 68.75% | 69.19% | 74.19% |
| central division makes such decisions | 1.03% | 2.17% | 14.59% | 25.81% |
| another person | 20.62% | 14.95% | 0.54% | 0.00% |

Source: own elaboration.

**Table 5.** Correlation between: the plans of company to use e-work after the SARS-CoV-2 pandemic and size of company.

| | Micro Companies (less than 10 Employees) | Small Companies (10–49 Employees) | Medium Companies (50–250 Employees) | Big-Sized Companies (over 250 Employees) |
|---|---|---|---|---|
| yes | 19.4% | 48.9% | 87.0% | 72.4% |
| no | 80.6% | 51.1% | 13.0% | 27.6% |

Source: own elaboration.

In the studied organizations, decisions on the implementation of innovations in most companies regardless of their size are made by the management of the company. This is the case in small (68.75%), medium (69.19%) and large organizations (74.19%). Micro-enterprises are an exception, where the decision-making role belongs to the manager (62.89%)—Table 4.

The opinion on the usability of telework after COVID-19 pandemic shifts with the size of the company. The businesses with 50 or more employees declare that they will use this form of employment in their further activities. In contrast, microenterprises strongly stand by their position (80.6%) that they will not give preference to remote working once the pandemic danger is over—Table 5.

## 6. Statistical Verification of Hypotheses

All three hypotheses formulated are statistical hypotheses, i.e.,: " . . . judgments on the general population, without full knowledge of those populations" [56]. In addition, they are parametric, meaning they are focused and related to population parameters. The hypotheses will be verified using a statistical test. The result of statistical tests performed will not be whether the hypotheses are true or false, but whether the researcher can accept or reject them in favor of accepting the alternative hypothesis and accepting the subjectively accepted test significance level $\alpha$. [57].

Statistical hypothesis tests are associated with the need to formulate a null hypothesis H0 and an alternative hypothesis H1. In some cases, more than one alternative hypothesis can be formulated. Making subjective assumptions—the test significance level—involves the risk of two types of errors. These are referred to as errors of the first and second type. [58].

An error of the first type is made in the situation of rejecting the null hypothesis $H_0$, which in fact is true. The probability of this error occurring is referred to as the test significance level and is denoted by the symbol $\alpha$. A second type of error occurs when the alternative hypothesis, $H_1$, is accepted as false. The probability of its occurrence is denoted by the symbol $\beta$ [59].

For all three hypotheses, some specific reference values should be adopted. Given the formulation and structure of H1, H2, and H3, it should be stated that they will be accepted

if the null hypotheses are rejected in favor of accepting the alternative hypotheses. An additional special condition related to the logic of adopted thesis H2 is that it can only be verified if $H1_2$ is accepted.

For testing all three hypotheses, a significance level of $\alpha = 0.05$ is assumed. The initial verification parameters are provided in Table 6.

**Table 6.** Parameters of statistical hypothesis verification.

| hypothesis | H1 | H2 | H3 |
|---|---|---|---|
| null hypothesis | $H1_0$ | $H2_0$ | $H3_0$ |
| alternative hypothesis | $H1_1$ | $H2_1$ | $H3_1$ |
| test significance level | $\alpha = 0.05$ | $\alpha = 0.05$ | $\alpha = 0.05$ |
| statistical test | Structure indicator test | Structure indicator test | Structure indicator test |

Source: own elaboration.

The population reference (*p*-value) was 0.5 (because H1 states about the majority of T&L companies, and that majority is more than 50% or more than 0.5). Verification of H1 required additional aggregate coding of responses and assigning them to specific ranges. It was decided to categorize the "definitely yes" and "yes" responses into the group consistent with H1, and the rest, including "don't know" response into the group inconsistent with H1. The null hypothesis took a form of: $H1_0$: $p = 0.5$. The alternative hypothesis took a form of: $H1_1$: $p > 0.5$. The right critical area was considered, as shown in Figure 10. The value of $K_1$ is the critical value for the test.

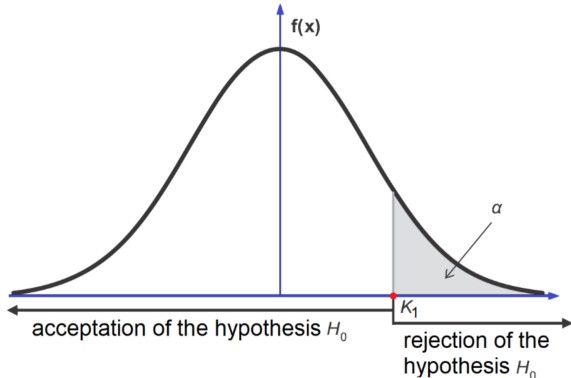

**Figure 10.** Graph of the right critical area. Source: own elaboration.

The right critical area was as follows: $K \leq K_1; +\infty)$, $K_1 = 1 - \alpha = 0.95 \rightarrow$ after reading the value of the distribution from the statistical tables $\rightarrow K_1 = 1.65 \rightarrow K \leq 1.65; +\infty)$. The chosen test statistic (statistical test) is the structure index test, which is expressed by the equation (Equation (2)):

$$U = \frac{\frac{m}{n} - p_0}{\sqrt{\frac{p_0 \times (1 - p_0)}{n}}} , \tag{2}$$

where:

$U$—structure indicator test,

$m/n$—structure indicator from the sample,

$p_0$—population reference value,

$n$—sample value,

$m$—number of elements distinguished in the sample.

After performing the calculations, the value of $U = 14.19$ was obtained. $U = 14.19 \in$ <1.65;+∞) → $U$ belongs to the critical area!

There is a statistical basis for rejecting $H1_0$ in favor of accepting $H1_1$, which is consistent with the general hypothesis H1 posed in this paper. As there is no statistical reason to disqualify H1, it will be verified on its merits.

The critical condition stating that H2 can only be accepted for verification if H1 is not rejected (and thus $H1_0$ is rejected in favor of $H1_1$) was met. The population reference value ($p$) was 0.5 (the hypothesis stated that the majority of T&L companies do not invest in innovation during a pandemic crisis, and as with H1, the majority is more than 50% or 0.5 of the total fraction). The null hypothesis had the following form: $H2_0$: $p = 0.5$. The alternative hypothesis was as follows: $H2_1$: $p > 0.5$. The right critical area was considered. $K \leq K_2$;+∞), $K_2 = 1 - \alpha = 0.95 \rightarrow K_2 = 1.65 \rightarrow K \leq 1.65$;+∞). The structure index test was: $U = 3.12$. $U = 3.12 \epsilon < 1.65$;+∞) $\rightarrow U$ belongs to the critical area!

This means that $H2_0$ should be rejected in favor of adopting $H2_1$, which is consistent with H2, thus there are no statistically significant reasons to not consider hypothesis H2.

H3 states that during the pandemic, the most common innovations in T&L companies were organizational innovations. In this case, the second highest score, i.e., the number of declarations of implemented process innovations, was taken as the reference value. Finally, the reference value ($p$) was 0.178. The hypotheses were presented as follows: null hypothesis: $H3_0$: $p = 0.178$ and alternative hypothesis: $H3_1$: $p > 0.178$. The right critical area was considered. $K \leq K_3$;+∞), $K_3 = 1 - \alpha = 0.95 \rightarrow K_3 = 1.65 \rightarrow K \leq 1.65$;+∞). The structure index test was: $U = 31.02$.

$U = 31.02 \epsilon < 1.65$;+∞) $\rightarrow U$ belongs to the critical area!

The interpretation of obtained result leads to rejection of $H3_0$ in favor of adopting $H3_1$, which is consistent with H3. Thus, there are no grounds for eliminating H3 and it can be considered further.

## 7. Conclusions

The COVID-19 pandemic changed the economic world and disturbed the functioning of markets. Unexpectedly, it also had a positive impact on, e.g., international scientific cooperation. Although it primarily concerned the medical sector, it is noteworthy that it brought together many countries and scientific institutions. However, not every scientific sector benefited from the pandemic. Scientific R&D projects had to be temporarily suspended due to the restrictions imposed. Therefore, it is extremely important to resume abandoned projects in the post-pandemic period as soon as possible and to support and protect all development and innovative activities [14,60].

The authors faced numerous problems and limitations while conducting the study. One of them was the need to use different types and formats of questionnaires, which significantly increased the scope of work. Another problem was the lack of funding for the research project, which deprived the authors of an external source of funding and thus limited the scope of study to three countries. The adoption of such a large research sample provides the opportunity to obtain very valuable data, but its processing and physical acquisition of respondents are very labor intensive for such a small research team. It should be noted that the study was conducted at a "critical" time with regard to the pandemic's development, and it seems impossible to reproduce it with such a high degree of reliability. This makes the results all the more valuable and unique. Due to the intention to reach as many entities as possible, the authors maximally simplified the survey questionnaire, focusing mainly on issues closely related to the implementation of innovation. As a result, the respondents were more willing to provide answers and the questionnaires were characterized by a high level of completeness and understanding of the issues addressed. Unfortunately, such an approach also had its drawbacks, such as the lack of questions about the economic aspects of implemented innovations, which in turn significantly impaired the possibility of conducting statistical analyses.

Another limitation was the use of subjectively selected advanced statistical tools and their limited number. There is a chance that conducting more in-depth analyses would

allow additional conclusions and insights to be drawn. Perhaps such activities will be conducted in the future and presented in future papers.

The completeness of results and a large number of comparative materials would certainly provide cyclical monitoring of changes occurring in the studied sectors, because it should be assumed that the motivations and actions of companies changed with the pandemic situation as well as regulations and restrictions. The crisis appeared suddenly and unexpectedly, hence many opinions were irretrievably lost, as questions about the distant past are generally subject to great uncertainty and error.

The authors of this paper attempted to illustrate innovation among T&L companies in Sweden, Poland and Germany during the global coronavirus pandemic. They also adopted three research hypotheses H1, H2 and H3. The H1 and H2 hypothesis should be considered true. This is proved by the research results, from which it can be concluded that the majority of companies (53.9%) decided not to introduce innovations during the pandemic, despite the fact that they were perfectly aware that this is a critical time for such actions (Figure 9), and companies that will make such an effort at the given time may achieve many benefits, especially in the period after the crisis. However, the fear of disease, lack of financial resources and expertise, lack of properly qualified employees and insufficient system support prevailed.

The second hypothesis was that during the COVID-19 pandemic the most frequently implemented innovations were organizational innovations. The basis for its formulation was the need to reorganize the company's operations resulting from the introduction of new legal regulations and the use of remote operation mode. The H2 hypothesis was also positively verified. It turned out that as much as 55% of innovations introduced in the T&L sector were organizational solutions.

In general, the situation in the T&L sector does not look good. There is a very large disparity between the observations made in different countries. The developing countries are still a long way from the highly developed ones, both in terms of innovation levels as well as awareness and perception of innovation needs. The lack of cohesion also occurs in the case of company size. The authors observed an interesting correlation, i.e., the bigger the company, the more open it is and more willingly reaches out and adopts new solutions and techniques.

The T&L constitutes a bloodstream of the EU economy. It employs about 20 million people, that is 12% of the total number of jobs available in Europe [61]. As every branch of economy is linked to GDP, and regardless of the simulation type, the forecasts for the coming years are not the best (Figure 2). However, in terms of innovation, in most cases companies bear implementation costs themselves, which gives them some independence and autonomy. However, this autonomy may turn out to be only an illusion, because the general economic downturn means a reduction of companies' profits and consequently investment resources. In addition, all innovations and R&D works show a very close dependence on state support, which is currently focused on the medical industry.

The results obtained are consistent with other studies conducted in Europe. The pandemic-induced economic crisis shook supply chains, which proved to be completely unprepared and inflexible in the context of this situation. The main reasons are considered to be the insufficient diversification characterizing logistics networks in terms of manufacturers, and the insufficient digital innovation level in the logistics sector [62].

According to the authors, innovations are an imperative that can be an effective response to the economic devastation caused by the global pandemic. In the T&L sector, not only the aspects related to new technologies, but also the rules of running a business have to be transformed, thus organizational, product and process innovations have to go hand in hand with technological innovations. The problem is multidimensionality and complexity of the T&L issue, because its different elements are at a different development level, have different priorities as well as different flexibility and ease of change. The key is a policy of sustainability, i.e., to ensure equal and green growth based on innovation.

The conducted consideration on the problem leads the authors to formulate three basic directions of activities that could effectively stimulate pro-innovative initiatives in the TSL sector.

1.  Observed production narrowing and an attempt to eliminate warehousing costs for goods during the COVID-19 crisis have left companies unable to meet orders. The directions of change in the logistics sector are not bad, but they are based on the assumption of effective profiling of the target customer. This is supported by the expansion of digital services and the development of information societies. However, there is a failure in the effectiveness and dimension of the application of new business intelligence analytical tools. Companies are too slow to adopt new solutions, which means that shortening the distance between the producer and the recipient is not fully effective.

2.  Another remedy for similar economic crises is to expand and increase the flexibility of logistics networks, not in terms of infrastructure, but in the number of partners in European countries.

3.  The research shows that entrepreneurs have high information awareness. They understand that innovations lead to progress and increased competitiveness on the market, but at the same time they are afraid of losing financial resources that would guarantee their survival in case of market collapse. In this case, the role of large international organizations such as the EU is very important, which should pay attention to the possibility of securing and guaranteeing companies that follow the path of innovation in research and development [21]. On a smaller scale, individual national governments should also stimulate such policies, while the research shows that only 50% of the countries make intensive use of innovation policies with external organizations (31.8% as supporters of cooperation and 18.2% as open collaborators) [33,63].

The authors plan to continue their research in the subject area. They plan to expand the research to three more countries: Germany, Denmark, and Lithuania. They assume to conduct two more rounds of surveys: at the very end of the pandemic and about a year after its end. This will allow for the collection of a very comprehensive comparative material and the creation of an in-depth report that will be made available for free online to all interested parties and organizations. To meet this challenge, the authors are currently in the process of preparing an application for funding of the described project from the program: Interreg Południowy Bałtyk 2021–2027. If the application proves successful, it is anticipated that two more scientific papers will be developed as a continuation of this dissertation.

**Author Contributions:** Conceptualization, M.K., E.G. and P.G.; formal analysis, M.K., E.G. and P.G.; methodology, M.K., E.G. and P.G.; writing and editing, M.K., E.G. and P.G.; visualization, M.K., E.G. and P.G. All authors have read and agreed to the published version of the manuscript.

**Funding:** This research did not receive any specific grant from funding agencies in the public, commercial, or not-for-profit sectors.

**Institutional Review Board Statement:** Ethical review and approval were waived for this study, due to the fact that this research design has discussed the perspective/opinion of the workers of different companies without any intervention to their individual action/behavior, and this research is not about the respondents themselves. This research is focused on the implementation of innovations in companies during the COVID-19 pandemic.

**Informed Consent Statement:** Not applicable.

**Data Availability Statement:** To receive a file in *.xls format, please send an e-mail to piotr.gutowski@usz.edu.pl.

**Conflicts of Interest:** The authors declare no conflict of interest.

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
