# Peer review of "Innovations in the T&L (Transport and Logistics) Sector during the COVID-19 Pandemic in Sweden, Germany and Poland"

_sustainability, doi:10.3390/su14063323_

Round 1
Reviewer 1 Report
Dear Author(s), I would like to thank you for the opportunity to read your manuscript entitled “Innovations in the T&L (transport and logistics) sector during the Covid-19 pandemic in Sweden, Germany and Poland”.
The overall manuscript is well presented with minor spelling or grammar mistakes, especially those with an inappropriate article used.
The overall work is very interesting, as the problem of recession in the field of transportation and logistics after the Covid-19 pandemic is very relevant.
Here are some issues concerning your paper:
- The overall purpose of the article should be stated clearly in the introduction and underlined in the abstract.
- The Literature Review part is insufficient. Have similar studies been carried out on the example of other countries? What were the main achievements of the research in previous years? The knowledge gap should be indicated.
- Figure 1: in the 55% there is “to low” – perhaps should be “too low” – I find it very colloquial. Consider other forms: weak skills, poor level etc. What are the “Needs and threats” indicated here? It looks like the shortages and weaknesses of T&L companies.
- Figure 2 needs more explanation. What are the Q1-Q4 next to the years? The unit of global GDP value on the OY axis (dollars?) should also be written in the figure.
- Table 1: consider changing the “I don’t know answer” into difficult to define or similar.
- Line 429: “Table 31. 7% of responses”
- Figure 9: definitely is spelled with mistake twice.
- In the final part, explain your findings according to the previous research on the topic.
- How are the determining indicators of the T&L companies' performance related to other countries of the world?
- Future research directions regarding the analyzed issues should be underlined and explained.
Reviewer
Author Response
Dear reviewer, we have made the following improvements to the text with regard to your comments.
- A clearly stated objective was included in the abstract and in the introduction. The following paragraph has been added:
The authors have focused on the research topic addressed as it is extremely relevant to the global economy as a whole. The size of the global logistics market may be indicated by the fact that it is expected to nearly double in value from just over $7.5 trillion in 2017 to nearly $13 trillion in 2027 in just a decade [46]. This is because there is an obvious convergence between the logistics market and the ICT sector and e-commerce in particular. Such a state of affairs results in accelerating the development of logistics while maintaining the principles of its sustainability becomes one of the main global imperatives. One of the factors influencing this acceleration is the effective adoption of innovations. The authors adopted as the aim of their research to investigate how this adoption progresses during the economic downturn caused by the omnipresent pandemic threat. The results and conclusions of conducted research may be very relevant and useful when implementing countermeasures in case of similar crises and provide a guideline for T&L companies and enterprises.
[46] M.Halmare, A.Jawarkar (2021), Report: Logistics Market by Mode of Transport (Railways, Airways, Roadways, and Waterways) and End Use (Healthcare, Manufacturing, Aerospace, Telecommunication, Government & Public Utilities, Banking & Financial Services, Retail, Media & Entertainment, Technology, Trade & Transportation, and Others): Global Opportunity Analysis and Industry Forecast, 2017–2027, Allied Market Research.
- The authors agree with the reviewer’s comment. The paper has been expanded by adding a literature review subsection.
The pandemic time, which is certainly a tragic and traumatic period in the history of modern civilization, had a very strong impact and influence on economic life. Many researchers and economists have taken advantage of this difficult time to conduct research aimed at creating more resilient and flexible economic models based on innovation, which can be seen as a weapon to fight COVID-19 [48], [49], [50]. Research in this context has also focused on the T&L sector, but has addressed its different aspects and focused on different regions of the world (Table 1).
Table 1. Examples of studies on the T&L sector during the COVID-19 pandemic
|
Title |
Study details |
Key findings |
|
COVID-19 Impact on the Logistics Industry [51] |
Surveyed entity: a company based in Dubai that offers its services and trades throughout the Middle East |
The negative impact of COVID-19 was observed in almost every aspect of the company’s operations. The company could not meet timely deliveries which ultimately led to a decline in its revenue |
|
Impact of COVID-19 on the supply chain industry [52] |
Impact of COVID-19 on the health of supply chains using Nigeria as an example |
The manufacturers were not able to meet the market demands due to inability to purchase raw materials on time. This situation led to increased inflation and depleted supply of goods. In order to improve the situation, some of the companies reached for innovative solutions usually based on ICT, which proved to be an effective solution |
|
Impact of COVID-19 on transportation and logistics: a case of China [53] |
Quantitative research on the impact of pandemic on T&L sector in China. The research focused on three spaces: land, sea and air logistics |
The study results proved that Covid-19 significantly and negatively affected the air and land logistics sector. However, no statistically significant relationship was observed with the maritime logistics sector |
|
Moving towards “mobile warehouse”: Last-mile logistics during COVID-19 and beyond [54] |
Literature study and analytical models |
Innovation related to the derivation of mobile storage can be an effective tool to counteract the difficulties caused by COVID-19 |
|
The Impact Of Covid-19 On Logistic Systems: An Italian Case Study [55] |
Case Study – quantitative and qualitative study of companies operating in the logistics industry in Italy |
All activities related to the introduction by companies of protections against COVID-19 had a significant impact on its financial health. During the pandemic, customer preferences changed which also affected logistics companies |
|
Digital Transformation in Latin American and Caribbean Logistics [56] |
Secondary data, literature review. South America and the Caribbean |
The Covid-19 pandemic became a catalyst for the digitization of trade and logistics |
|
[57] Fast Forward. Rethinking supply chain resilience for a post-COVID-19 world |
Quantitative and qualitative research. 1.000 surveys targeted to consumer industry entrepreneurs and in-depth interviews with selected retail chain executives. The survey was conducted in 11 selected countries around the world |
The supply chains of the vast majority of organizations did not survive the test of coronavirus pandemic. Based on this experience, companies have taken numerous initiatives to increase their supply chain flexibility, but they do so on sustainable terms |
Source: Own elaboration
It is important to note that the main goal of T&L companies is to store and distribute goods efficiently through flexible supply chains [58]. However, the emergence of the pandemic threat has put the entire logistics system to a very demanding test [59]. Operators have comprehended an uneven fight against numerous constraints (such as restrictions) and other problems (e.g., staff shortages or availability of goods and services) in order to maintain the entirety of meeting their customers’ needs [60], [61]. Some researchers have realized that innovation is the most effective way to achieve this goal and have focused their interest and research potential on it [62]. Despite the relatively large number of publications in the sector, according to the authors, there is still a lack of sufficiently insightful primary research, which is not a case study of one company (such as [63]), one country (such as [64]) or based on secondary data (such as [65]). It should be emphasized that the conclusions presented in the paper are based on research conducted in three countries and concerning as many as 1597 economic entities, which is very rare in scientific papers. This makes the results obtained extremely valuable and have a very high credibility level. However, it cannot be ruled out that similar studies will appear in the near future, as the pandemic is a phenomenon that began suddenly and is still ongoing.
[48] Report UN Innovation Network: Innovation Covid-19 Special Edition (2020), https://www.un.org/sites/un2.un.org/files/2020-06_–_unin_quarterly_innovation_update_–_second_covid-19_special_edition.pdf
[49] B.Ramalingam (2020), Innovation, development and COVID-19: Challenges, opportunities and ways forward, OECD Policy Responses to Coronavirus (COVID-19), p. 2.
[50] H.Orkibi (2021), Creative Adaptability: Conceptual Framework, Measurement, and Outcomes in Times of Crisis. Front. Psychol. 11:588172. doi: 10.3389/fpsyg.2020.588172
[51] D.S. Siddiqui (2020), COVID-19 Impact on the Logistics Industry, A case study developed with a social and economic sustainability perspective on a firm operating in the middle east, Jönköping University, p. 45.
[52] PWC report (2020), Impact of COVID-19 on the supply chain industry, p. 15.
[53] Y.Xu, J.P.Li, C.C.Chu, G.Dinca (2021), Impact of COVID-19 on transportation and logistics: a case of China, Taylor & Francis, Economic Research-Ekonomska Istraživanja, DOI: 0.1080/1331677X.2021.1947339
[54] S.Srinivas, R.R.Maratheb (2021), Moving towards “mobile warehouse”: Last-mile logistics during COVID-19 and beyond, Transportation Research Interdisciplinary Perspectives, Volume 10, ISSN 2590-1982, DOI: https://doi.org/10.1016/j.trip.2021.100339
[55] M.Rinaldi, T.Murino, E.Bottani (2021), The Impact Of Covid-19 On Logistic Systems: An Italian Case Study, IFAC-PapersOnLine, Volume 54, Issue 1, ISSN 2405-8963, DOI: https://doi.org/10.1016/j.ifacol.2021.08.123
[56] Digital Transformation in Latin American and Caribbean Logistics (2020), in: FAL Facilitation Acilitation Cilitation of Transport and Trade in Latin America and The Caribbean, bulletyn 381, number 5/2020/ISSN: 1564-4227
[57]Report Capgemini (2020) ,Fast Forward. Rethinking supply chain resilience for a post-COVID-19 world, https://www.capgemini.com/wp-content/uploads/2020/11/Fast-forward_Report.pdf
[58] M.Umar, X.Ji, D.Kirikkaleli, Q.Xu (2020), Do innovation, financial development, and transportation infrastructure matter for environmental sustainability in China? Journal of Environmental Management, 271, https://doi.org/10.1016/j.jenvman.2020.111026
[59] L.Yarovaya, N.Mirza, J.Abaidi, A.Hasnaoui (2021), Human capital efficiency and equity funds’ performance during the COVID-19 pandemic. International Review of Economics & Finance, 71, https://doi.org/10.1016/j.iref.2020.09.017
[60] N.Mirza, J.A.Hasnaoui, B.Naqvi, S.K.A.Rizvi (2020), The impact of human capital efficiency on Latin American mutual funds during Covid-19 outbreak. Swiss Journal of Economics and Statistics, 156(1), p. 4
[61] Su, C.-W., Dai, K., Ullah, S., & Andlib, Z. (2021). COVID-19 pandemic and unemployment dynamics in European economies. Economic Research-Ekonomska Istraživanja, p. 1–13. https://doi.org/10.1080/1331677X.2021.1912627
[62] I.Dovbischuk (2021), Innovation-oriented dynamic capabilities of logistics service providers, dynamic resilience and firm performance during the COVID-19 pandemic, The International Journal of Logistics Management Emerald Publishing Limited, DOI: 10.1108/IJLM-01-2021-0059
[63] R.Wilding, K.Dohrmann, M.Wheatley, Report DHL, Post-Coronavirus Supply Chain Recovery. A DHL perspective on the impact of the COVID-19 pandemic on supply chains and Logistics.
[64] Report Government of Ireland, Department of Business, Enterprise and Innovation (2020), Focus on Transport and Logistics, https://enterprise.gov.ie/en/Publications/Publication-files/Focus-on-Transport-and-Logistics-2020.pdf
[65] A.Parfenov, L.Shamina, J.Niu, V.Yadykin (2021), Transformation of Distribution Logistics Management in the Digitalization of the Economy. J.Open Innov. Technol. Mark. Complex., 7, 58, DOI: https://doi.org/10.3390/joitmc7010058
- The authors agree with the reviewer’s comments. The figure and its caption have been changed.
- In Figure 2, the unit is %. The reference point is Q4 2019 and has been assimilated to 100%. Further results and changes are defined relative to this reference value. The figure has been changed according to the reviewer’s comment – the % designation has been added.
The presented forecast covers a specific period of time and it should be mentioned that it is rather illustrative, because the pandemic effects are very turbulent and the situation changes very dynamically. The source from which the figure is derived also does not contain information and forecasts for the future. In the authors’ opinion, speculating in the more distant future is completely unnecessary as it may be subject to a very large error. Therefore, the authors have abandoned further analysis of Figure 2 as suggested by the reviewer.
- The entry in Table 1 has been changed as suggested by the reviewer.
- An error in the text has been removed.
- Errors in Figure 9 have been corrected according to the reviewer’s comment.
- According to the authors, the summary is complete and sufficiently elaborated references to the issues addressed have been included in the paper.
- During their research and literature studies, the authors noted a very high level of consistency between their observations and studies conducted in other countries. This is highlighted in many places in the paper. An example is H1, which was based on research by the McKinsey Institute and was found to be true. Given the timeliness of the subject addressed, the authors expect more reliable data to become available in the near future, which they plan to use for comparisons in future publications. This elaboration is addressed to the reviewer only, as the authors believe there is no need to include it in the paper.
- The authors agree with the reviewer’s suggestion. The following notation was included in the conclusion:
The authors plan to continue their research in the subject area. They plan to expand the research to three more countries: Germany, Denmark, and Lithuania. They assume to conduct two more rounds of surveys: at the very end of the pandemic and about a year after its end. This will allow for the collection of a very comprehensive comparative material and the creation of an in-depth report that will be made available for free online to all interested parties and organizations. To meet this challenge, the authors are currently in the process of preparing an application for funding of the described project from the program: Interreg Południowy Bałtyk 2021-2027. If the application proves successful, it is anticipated that two more scientific papers will be developed as a continuation of this dissertation.
Dear Reviewer. The authors would like to thank you very much for your constructive comments, which will certainly allow us to publish more correct scientific papers in the future.
Sincerely, Authors
Reviewer 2 Report
This is an interesting paper. It has some compelling analysis. However, some improvements are required, as per below:
- The abstract is too vague. It needs to be rewritten to include some of the actual findings.
- The keywords are also too general. consider using more specific keywords.
- Although the introduction section does touch on the aims, however, this needs to be comprehensively deliberated.
- The methodology also needs to be revised to explain why the selected method was used. Why other methods such as differential equations were not considered etc.
- Table 2 needs more deliberation since it is the centerpiece for the findings.
- The conclusion section also needs to discuss some of the limitations of the research. Remember that the paper is presented to a global audience and as such need to consider the broader context.
Finally, there are some minor grammar issues that require editing.
Author Response
Dear reviewer, we have made the following improvements to the text with regard to your comments.
- The abstract has been expanded to present the paper’s content more clearly. It now reads as follows:
Abstract: Innovation is one of the most important factors stimulating the economy. It plays a special role in the T&L sector as it enables the acceleration of meeting needs process. During the COVID-19 coronavirus pandemic, many industries were and still are facing a tough economic test. The recession is also noticeable in transport, freight forwarding and logistics. But how does this sector cope with the existing problems? Has the adoption rate of innovation been stopped in this sector? Do T&L developers see the potential of innovations and do they see them as a remedy and response to the pandemic threat? These issues have been thoroughly considered in the presented publication.
The paper presents conclusions and selected results from a study on the adoption of innovations by companies in the transport and logistics sector during the COVID-19 coronavirus pandemic in Sweden, Germany and Poland. As many as three research hypotheses were adopted, which after being subjected to statistical fractional verification and evaluated substantively on the basis of the literature review and conclusions of research conducted, proved to be true. The aim of this paper was to verify the principles and determinants of innovation policy in T&L enterprises in selected countries during the pandemic crisis. Moreover, the paper contains an analysis of the entrepreneurs’ experiences in the context of improving and developing their activities during economic crises, e.g. in 2008. It also presents the motivation and methodology of research. In addition to standard quantitative summaries, the authors conducted identification of correlations between the studied phenomena using the Cramer’s V method and chi-square statistics.
Obtained results allowed to better understand the processes taking place and to determine the general state and prospects of further innovation development in the T&L sector during the pandemic and ubiquitous restrictions.
- The keywords have been revised to be more specific. They are now presented as follows: innovation during the crisis, innovation in the T&L sector, statistical fractional verification, statistical correlations.
- The authors take the position that the rationale for the objectives of the study and the subject matter undertaken has been sufficiently argued; however, in order to improve the clarity of this publication, they have decided to the following paragraph:
The authors have focused on the research topic addressed as it is extremely relevant to the global economy as a whole. The size of the global logistics market may be indicated by the fact that it is expected to nearly double in value from just over $7.5 trillion in 2017 to nearly $13 trillion in 2027 in just a decade [46]. This is because there is an obvious convergence between the logistics market and the ICT sector and e-commerce in particular. Such a state of affairs results in accelerating the development of logistics while maintaining the principles of its sustainability becomes one of the main global imperatives. One of the factors influencing this acceleration is the effective adoption of innovations. The authors adopted as the aim of their research to investigate how this adoption progresses during the economic downturn caused by the omnipresent pandemic threat. The results and conclusions of conducted research may be very relevant and useful when implementing countermeasures in case of similar crises and provide a guideline for T&L companies and enterprises.
[46] M.Halmare, A.Jawarkar (2021), Report: Logistics Market by Mode of Transport (Railways, Airways, Roadways, and Waterways) and End Use (Healthcare, Manufacturing, Aerospace, Telecommunication, Government & Public Utilities, Banking & Financial Services, Retail, Media & Entertainment, Technology, Trade & Transportation, and Others): Global Opportunity Analysis and Industry Forecast, 2017–2027, Allied Market Research.
- The methodology of this study has been thoroughly considered. The authors, having experience in conducting research involving the identification of hidden statistical relationships between phenomena in quantitative research, but focusing on qualitative features. In this regard, the number of statistical tools is limited. In such a situation, the authors most often resort to the Pearson correlation coefficient and Cramer’s V method, although the latter, especially in combination with the chi-square statistic, provides more objective results (conclusion based on experience and comparisons). Regarding the reviewer’s comment, the authors are not sure exactly what the reviewer meant. Differential equations appear in several methods and it is not specified exactly which one is meant…?
- The authors have developed an interpretation of the results in Table 2. However, they do not interpret it as the key point of the paper, because the conducted research is so novel, unique and valuable that the conclusions from their basic aggregation and comparison alone are, in the authors’ opinion, interesting enough. The correlation analysis was carried out as an additional activity, nevertheless, due to the reviewer’s opinion, the text was supplemented.
The added section is as follows: The strongest correlation was found between the motivation for innovation and the company’s country of origin. The specific results are presented in Figure 6. In this case, the value of the Cramer’s V coefficient was as high as 0.673, which is the highest result in the entire study. Unfortunately, the method used does not allow to determine the directionality of the identified relationship, but only to give an opinion on its strength, which in this case should be described as very high.
While in the case of Poland and Germany, the opinions of respondents were very similar, in the case of Sweden the economic motivation was by far the most important. Looking at constantly updated data on an interactive map developed by the Johns Hopkins University [47] concerning COVID-19 pandemic, it is possible to easily notice the huge disproportion in relation to the number of deaths in the discussed countries. Taking into account the status as of 26.02.2022, in Germany there were 122 639, in Poland 111 277 and in Sweden 17 142 victims of Coronavirus. Due to the differences in population, it is worth introducing a simple auxiliary indicator (T) equal to the ratio of population and pandemic deaths. After performing the calculations, it was found that: TSE=603, TDE=679, TPL=341. The results obtained should be interpreted as follows: by the end of March 2022, one person out of every 603 citizens has died due to COVID-19 in Sweden, one person out of every 679 citizens in Germany and one person out of only 341 citizens in Poland. Poland’s performance is by far the most unfavorable. The situation was almost twice as good in Sweden and Germany, respectively. The analysis performed is consistent with the study results. It was in Poland that the pandemic was the main motivator (35.5% – Figure 6) for the introduction of innovations, many of which involved innovative ways of protecting life and health.
[47] Coronavirus COVID-19 Global Cases by John Hopkins University, https://gisanddata.maps.arcgis.com/apps/dashboards/bda7594740fd40299423467b48e9ecf6
- The authors write about the study’s limitations in the second paragraph of the summary. However, due to the reviewer’s comment, this paragraph was expanded to include the notation:
Another limitation was the use of subjectively selected advanced statistical tools and their limited number. There is a chance that conducting more in-depth analyses would allow additional conclusions and insights to be drawn. Perhaps such activities will be conducted in the future and presented in future papers.
The completeness of results and a large number of comparative material would certainly provide cyclical monitoring of changes occurring in the studied sectors, because it should be assumed that the motivations and actions of companies changed with the pandemic situation as well as regulations and restrictions. The crisis appeared suddenly and unexpectedly, hence many opinions were irretrievably lost, as questions about the distant past are generally subject to great uncertainty and error.
- In accordance with the reviewer’s comment, the text was subjected to analysis and editorial revision. The result was the removal of several errors and linguistic awkwardness.
Dear Reviewer. The authors would like to thank you very much for your constructive comments, which will certainly allow us to publish more correct scientific papers in the future.
Sincerely
Authors
Round 2
Reviewer 1 Report
Dear Authors, thank you for adapting the text of the article to my recommendations. I strongly believe that the improvements introduced have a big impact on the transparency and logic of the presented research material. I wish you success in further research.
Regards,
Reviewer
This manuscript is a resubmission of an earlier submission. The following is a list of the peer review reports and author responses from that submission.
Round 1
Reviewer 1 Report
Dear authors,
The article to my way of seeing is interesting, however, I think it should be improved in the following aspects:
1. The introduction does not explain any motivation why the reader have to read the paper.
2. The approach carried out in this work is very elementary.
3. Comments of the survey are too general. I wonder that there is no acknowledgment of results by previous studies, try to add some validation of your results with the literature backup.
4. Literature review is poor and treats generally the research topic.
5. In the discussion the analysis should be reinforced (and if possible the comparison) with other cases of countries where the phenomenon was studied. Some statements in the document appear as subjective opinions that need to be supported in the literature.
6. Implications could and should be better explored and developed.